# OpenReview forum: "Rethinking Bottlenecks in Safety Fine-Tuning of Vision Language Models"
_ICLR.cc/2026/Conference — ICLR 2026 Poster_

### Official Review · Reviewer_gNSB · 2025-10-20

**Soundness:** 3
**Presentation:** 3
**Contribution:** 2
**Rating:** 6
**Confidence:** 4

**Summary:**

The paper studies a more complex safety setup than considered before. There are two images and a text given to an MLLM in order to jailbreak it. The idea is that on their own, both images and the text would be without safety concerns, but when reasoning is applied, the combination is unsafe. The paper introduces a dataset that is used for both fine-tuning of pre-trained MLLMs and also evaluating them. Experimental evaluation shows that existing models can be easily jailbroken using the data, but a model fine-tuned using the data is resistant to these kinds of attacks.

**Strengths:**

•	The paper focuses on a new safety problem setup where the combination of multiple images and a text is unsafe. This is a practically important problem setting as existing models cannot defend themselves well against it and the attack success rate is high.

•	The main contribution of the paper is a dataset that can be used for both fine-tuning of MLLMs and evaluating them for the described form of reasoning-based attack. The pipeline for creating the dataset consists of four stages and seems appropriate for the task. The dataset consists of a few thousand examples.

•	The experimental evaluation shows that existing models fail on the considered problem setting. Fine-tuning on the proposed dataset helps defend against such attacks, and it also helps for general performance. The evaluation uses a solid number of benchmarks for testing both general ability and defence against attacks.

•	A larger number of analyses is performed, and this is helpful for analysing the contribution and for understanding the overall problem setup better.

**Weaknesses:**

•	The described method MIRage appears to be essentially common fine-tuning on the proposed dataset, so we may not really see it as a specialized new method. Potentially we could say it is VLGuard with different data.

•	The main results seem to only use one model, even if there are some results on other models in the appendix. It would be good to see experiments with more models, e.g. in table 4 in the main text to have a better overview of how the solution works across multiple models.

Minor: small typos such as in L150 "Disccusions", I recommend proof-reading as there are a few more I recall, but not many

**Questions:**

What exactly is the difference between VLGuard-R and MIRage methods? Could MIRage also be seen as extension of VLGuard?

How computationally expensive was it to create the dataset?

---

> ### Author Response · Authors · 2025-11-18
>
> Thanks for your valuable feedback, which greatly contributed to improving the paper!
>
> ---
> > W1
> - **Failures of VLGuard**
> In Section 2, our analysis reveals that existing methods such as *VLGuard* lack the ability to reason about safety within visual contexts. This limitation leads to failures in balancing helpfulness and harmlessness, and difficulties in handling complex scenarios that extend beyond basic visual perception. While VLGuard aims to help models perceive what is safe or unsafe, we observe that this approach often results in overly conservative behavior toward visual content, hindering the model’s capacity to make nuanced, context-aware judgments.
>
> - **The Goal of MIS and MIRage**
> To address these challenges, our work introduces multi-image input data paired with safety CoT annotations. This design encourages models to perform safety-related cross-image visual reasoning, enabling them to infer and reason about safety rather than relying solely on surface-level perception. By enhancing visual reasoning, our approach improves both safety awareness and contextual understanding, achieving a better balance between helpfulness and harmlessness.
> - **A Novel yet Simple Dataset (MIS) to Resolve the Problem**
> Although we employ a standard SFT paradigm, our novel combination of multi-image inputs and safety CoT annotations effectively overcomes the bottlenecks of VLGuard and other baselines. This simple yet powerful dataset design substantially enhances the model’s safety performance by strengthening its visual reasoning capabilities.
> ---
> >W2
> - **We have presented MIRage on more models and evaluated it on the MIS dataset (Table 15)**
> Due to space limitations, we primarily reported the performance of MIRage on InternVL2.5-8B in the main text, including comparisons with other baselines and ablation studies. However, in the Appendix, we also provide comprehensive results of MIRage on *Qwen2-VL-7B-Instruct*, *MiniCPM-v2.6*, and *LLaVA-OneVision-7B*. Specifically, the complete results corresponding to your mentioned *Table 4* are presented in *Appendix Table 15*, which includes MIRage fine-tuned on these three additional VLMs as well as its inference-time defense performance on the MIS dataset.
>
> - **MIRage works effectively across diverse VLMs**
> Furthermore, Tables 13 and 14 report MIRage’s performance on multiple safety and general datasets across the same three models. These experiments collectively demonstrate that MIRage consistently achieves strong and stable improvements across diverse VLM architectures.
>
> We have merged the complete results from Tables 4 and 15 into a single table in the revision.
>
> ---
> > W3: small typos
>
> Thanks for your valuable suggestion. We have carefully reviewed our manuscript and corrected the typos accordingly.
>
> ---
> > Q1: What exactly is the difference between VLGuard-R and MIRage methods? Could MIRage also be seen as extension of VLGuard?
> - **The difference between VLGuard-R and VLGuard-P**
> VLGuard-P, introduced in the original VLGuard paper, contains images with explicitly harmful visual elements and uses rejection-style labels that begin with “I’m sorry,”. Our analysis shows that this training paradigm causes the resulting models to become overly conservative toward visual content. To address this, we reconstruct the labels in VLGuard-P using our Safety CoT–based labeling strategy, producing VLGuard-R. It is important to note that VLGuard-R is also proposed and constructed in our paper.
> - **The difference between VLGuard-R and MIRage**
> Motivated by the findings in Section 2, we aim to improve model safety through enhanced visual reasoning. However, we observe that the images used in VLGuard-R are too simple, making it difficult to produce high-quality safety CoT labels. To overcome this limitation, we introduce the MIS dataset, where harmful intentions naturally emerge through cross-image reasoning—by simply combining two individually safe images, a harmful scenario can be formed.
> As a result, MIRage and VLGuard-R differ both in their input formats and in the nature of their safety CoT labels, with MIRage involving significantly more complex cross-image visual reasoning.
> - **MIRage is not a simple extension of VLGuard**
> MIRage encourages models to perform safety-oriented cross-image reasoning, enabling them to infer safety risks instead of relying solely on surface-level perception—the primary target of VLGuard. By strengthening the model’s visual reasoning capabilities, MIRage improves both safety awareness and contextual understanding, achieving a more effective balance between helpfulness and harmlessness and addressing the bottlenecks identified in Section 2.
>
> ---
> > Q2: How computationally expensive was it to create the dataset?
>
> All experiments are conducted on a single machine with 8×A100 80GB GPUs. For data construction, we believe that using 4×A100 80GB GPUs would be more appropriate, as inference requires running 72B and 78B-sized models.

---

> > ### Comment · Reviewer_gNSB · 2025-11-23
> > **Thank you**
> >
> > Thank you for the additional explanations, I maintain my rating that I lean towards accepting the paper.

---

> ### Author Response · Authors · 2025-11-24
> **Thanks for the positive feedback**
>
> Thank you for your constructive suggestions and your response. We sincerely appreciate that our rebuttal has helped address all of your concerns. If there are any remaining questions or suggestions for improvement, we would be grateful to receive them.

---

### Official Review · Reviewer_v3WZ · 2025-10-28

**Soundness:** 2
**Presentation:** 2
**Contribution:** 3
**Rating:** 4
**Confidence:** 4

**Summary:**

The paper
- Demonstrates two Issues after safety finetuning with VLGuard data
    - Models are prone to overrefusal on benign data, reducing performance on general instruction following data (Table1, left)
    - Models struggle with complex safety situations where compositional dependencies between safe text and safe image can still lead to an unsafe request (Table1, right)
- To address these issues the paper
    - Constructs a new dataset (MIS) where each datapoint consists of text instructions referring to two images and the safety implications of the request requires reasoning about how the two images relate.
    - Generates safe responses for the constructed dataset by prompting InternVL2.5-78B with a general CoT prompt to reason about the inputs and provide a safe response.
    -  Finetunes multiple VL models with the generated training data
- Evaluation includes
    - multiple existing general VL instruction following and safety tasks.
    - A new multi-image safety test set from the newly constructed data
- Results show
    - Existing VLMs struggle with providing safe responses to the new multi-image safety data.
    - Models fine-tuned on the new MIS dataset perform almost perfectly on its test set.
    - Models fine-tuned on the new MIS dataset preserve general VL capabilities and improve safety on existing safety datasets.
    - Models fine-tune on the new MIS dataset outperform finetuning on VLGuard labels that were generated in the same way as the MIS labels.

**Strengths:**

- Construction of the Multi-image safety dataset seems like a valuable contribution especially since many existing models seem to struggle with providing safe responses to the inputs. It is important to be robust against adversarial instructions embedded in multiple images.
- Extensive experiments supporting that improvement from fine-tuning on the new data seems robust across base models and evaluation sets.
- VLGuard-R is a decent baseline and its substantially improved performance over existing methods is interesting in itself.

**Weaknesses:**

- I am somewhat confused by the results in table 4. How can InternVL2.5 78B perform so much worse than the proposed fine-tuning method if it provided the labels for that fine-tuning in the first place?
    - This suggests that the evaluation in table 4 does not prompt the models towards providing safe responses.
    - For a fair comparison, I would want to see the performance of those models when prompted with the same instructions as was done to generate the SFT labels.
    - This would give a more realistic sense of how impactful the whole dataset creation and fine-tuning procedure was that the paper proposes and to what extend the contribution is contained in the CoT prompt for the label generation.

I would be likely to support acceptance if this concern is adequately addressed and the claims are adjusted based on the results with the adapted prompt.

**Questions:**

- See the point specified in ‘Weaknesss’
- Is multi image necessary? Wouldn’t it be sufficient to train the model on more text-image compositional examples like those given in the MSS benchmark?

---

> ### Author Response · Authors · 2025-11-18
>
> Thanks for your valuable feedback, which greatly contributed to improving the paper!
>
> ---
> > W1: I am somewhat confused by the results in table 4. How can InternVL2.5 78B perform so much worse than the proposed fine-tuning method if it provided the labels for that fine-tuning in the first place?
> - **When constructing the safety CoT labels, we explicitly inform the model that the query contains safety risks in order to obtain fully safe responses.**
> As shown in Appendix H.2, the prompt we use during label construction is designed solely to obtain a fully safe response with a complete safety CoT. At this stage, we explicitly inform the model that the query contains safety risks and instruct it to first analyze the visual content, then identify potential safety hazards, and finally provide a safe answer. Under this setting, the model is able to generate fully safe and coherent responses. We also evaluated the training labels using GPT-4o, and the Safe Rate is 100%.
> - **Our evaluation protocol is fully consistent with existing safety benchmarks, and we do not provide the model with any hints about the safety risks contained in the questions.**
> Since safety benchmarks are designed to evaluate a model’s inherent safety capabilities, prior work does not disclose or hint at the safety risks contained in the input during testing; otherwise, the evaluation would not reflect the model’s true safety behavior. Our evaluation in Table 4 strictly follows this standard protocol, consistent with previous benchmarks such as MM-SafetyBench and VLSBench. Using the same prompt from the data-construction stage for evaluation would be unfair, as it would introduce data leakage—the model would already know in advance that the user query is harmful. In real-world usage, such prompts would also severely degrade general performance, causing the model to become overly conservative. Therefore, this prompting strategy is inappropriate for testing and does not reflect the model’s actual safety capability.
> - **As shown in Appendix Table 15, we evaluate the effect of using different CoT prompts during inference.**
> The Customized Safety CoT Prompt explicitly instructs the model to first analyze whether the query contains safety risks and then provide an answer. Although such a prompt can indeed reduce the ASR to some extent, it also **significantly harms** the model’s general performance. The table below reports the results of applying the Customized Safety CoT Prompt on general benchmarks. The results clearly demonstrate that this prompting strategy greatly reduces the model’s usefulness, confirming that relying on such prompts at inference time is not a practical or effective solution.
>
>    |Methods|Q-Bench (Acc ↑)|MuirBench (Acc ↑)|
>    |-|:-:|:-:|
>    |Qwen2-VL-7B-Instruct|77.32|40.77|
>    |+ Customized safety CoT|68.83|37.38|
>    |+ MIRage|77.93|42.31|
>
> - **Safety CoT label is important in MIRage fine-tuning**
> In Tables 11 and 12, we conduct ablation studies to examine whether the training labels include safety CoT. The results show that adding safety CoT labels significantly improves performance on SIUO and MSSBench, both of which require strong reasoning ability for the model to generate safe responses. In contrast, models fine-tuned without safety CoT labels exhibit a decline in general performance, whereas MIRage achieves consistent improvements across all five general benchmarks. This confirms that MIRage effectively gains stronger reasoning capability, which in turn enhances both safety and general performance.
>
> ---
>
> > Q2: Is multi image necessary? Wouldn’t it be sufficient to train the model on more text-image compositional examples like those given in the MSS benchmark?
>
> We believe that multi-image inputs are highly effective, though they may not be the only viable option.
> - **The key difference between MIS and our constructed VLGuard-R lies in the number of image inputs.** MIS uses multi-image inputs to create more complex cross-image reasoning scenarios, enabling the generation of higher-quality safety CoT labels. This, in turn, leads to stronger reasoning ability in the fine-tuned model.
> - **Although MSSBench also provides high-quality input data, its construction process is highly labor-intensive and difficult to scale.** In contrast, MIS is built through a fully automated pipeline with strong extensibility. Exploring how to use automated data construction frameworks to generate MSSBench-like high-quality datasets may be an interesting direction for future research.

---

> > ### Comment · Reviewer_v3WZ · 2025-11-26
> >
> > Thank you for the comprehensive response. One more follow up question:
> >
> > >  Since safety benchmarks are designed to evaluate a model’s inherent safety capabilities, prior work does not disclose or hint at the safety risks contained in the input during testing; otherwise, the evaluation would not reflect the model’s true safety behavior.
> >
> > I appreciate that this is common practice for academic benchmarks, though I disagree that "the evaluation would not reflect the model’s true safety behavior.". In practice, pretty much every model will be deployed with safety-specific instructions in its system prompt and thus it makes sense to at least evaluate that setting, which you have also done in the mentioned Table 15.
> >
> > I acknowledge that for the Qwen2-VL-7B-Instruct model, the Customized Safety CoT Prompt leads to a considerably worse operating point for the helpfulness-safety trade-off as you show in your response above.
> > My main question is: do you also observe the same trend for larger models, which should be more capable in instruction following?
> > E.g. in Table 15, what is the performance of Qwen2-VL-72B-Instruct with the Customized Safety CoT Prompt? And how would it impact helpfulness for these larger models? The same analysis for InternVL2.5 78B would also be very interesting as this is the model used for data generation.
> >
> > Finally, can you share the Vanilla CoT, Customized CoT and Customized Safety CoT prompts that you used for these evaluations?

---

> > > ### Author Response · Authors · 2025-11-29
> > >
> > > Thanks for your detailed reply.
> > > # Results on larger models
> > > We further applied the Customized Safety CoT prompt to larger models (Qwen2-VL-72B and InternVL2.5-78B), and the results are summarized in the tables below. Implicitly instructing the VLM to judge whether the input is safe does not reduce the ASR to an acceptable level, and it additionally degrades the model’s general performance. This trend on larger models is consistent with our observations on smaller models.
> > > Overall, these results indicate that prompt-based approaches are not an ideal solution for improving a model’s safety-related visual reasoning capabilities and tend to disrupt the balance between helpfulness and harmlessness.
> > > - **Results on MIS-test**
> > > |Models|easy-ASR(↓)|easy-HR(↓)|easy-RSR(↑)|easy-RR(↑)|hard-ASR(↓)|hard-HR(↓)|hard-RSR(↑)|hard-RR(↑)|real-ASR(↓)|real-HR(↓)|real-RSR(↑)|real-RR(↑)|
> > > |-|:-:|:-:|:-:|:-:|:-:|:-:|:-:|:-:|:-:|:-:|:-:|:-:|
> > > |Qwen2-VL-7B-Instruct|90.02|0.12|9.73|0.24|89.41|0.20|10.20|0.19|81.00|0.00|17.00|1.00|
> > > |+ Costumized Safety CoT|70.57|0.00|29.43|0.00|69.22|0.00|30.78|0.00|57.00|0.00|43.00|0.00|
> > > |Qwen2-VL-72B-Instruct|93.19|0.00|6.39|0.42|92.35|0.00|7.45|0.20|83.00|0.00|15.00|0.00|
> > > |+ Costumized Safety CoT|71.22|0.00|28.48|0.30|68.03|0.00|31.67|0.30|66.00|0.00|33.00|1.00|
> > > |InternVL2.5-78B|85.67|0.12|9.73|4.48|87.25|0.00|12.55|0.20|78.00|1.00|18.00|4.00|
> > > |+ Costumized Safety CoT|58.15|0.30|28.18|13.37|69.41|0.00|28.82|1.76|50.00|0.00|39.00|11.00|
> > >
> > > - **Results on general benchmarks**
> > > |Models|Q-Bench (Acc ↑)|MuirBench (Acc ↑)|
> > > |-|:-:|:-:|
> > > |Qwen2-VL-72B-Instruct|80.23|59.18|
> > > |+ Costumized Safety CoT|74.32|54.87|
> > > |Internvl2.5-78B|77.38|62.27|
> > > |+ Costumized Safety CoT|72.77|59.03|
> > >
> > > # Prompt we used in our paper
> > >
> > > - **Vanilla CoT:**
> > >
> > > Think it step by step.
> > >
> > > - **Customized CoT:**
> > >
> > > Think it in two steps: first, recognize the image; then, answer the question.
> > >
> > > - **Customized Safety CoT:**
> > >
> > > First, assess whether the user input is safe. If the inputs are safe, answer the question directly. If the
> > > inputs are not safe, generate a safety rule before answering the question.

---

### Official Review · Reviewer_PDkF · 2025-10-28

**Soundness:** 2
**Presentation:** 3
**Contribution:** 2
**Rating:** 6
**Confidence:** 3

**Summary:**

The paper identifies the bottlenecks in current safety fine-tuning methods, the helpful harmless trade-off, and their failures on challenging safety tasks. To address these issues, the authors introduce MIS, a dataset designed for reasoning-based safety understanding across multiple images, and propose MIRage, a fine-tuning method that enhances safety reasoning without harming general utility.

**Strengths:**

1. The paper systematically diagnoses and analyzes current safety fine-tuning bottlenecks through comprehensive experiments.
2. Authors present MIS, the first multi-image safety dataset, featuring a training split aimed at enhancing models’ safety-related visual perception and reasoning abilities.
3. Models finetuned with MIRage generalize well to previously unseen safety categories.

**Weaknesses:**

1. The “Safety CoT” template is interesting but not well formalized or analyzed—it’s unclear how much of the improvement comes from reasoning versus simply from data diversity. The authors should include ablations to isolate the impact of Safety CoT vs. multi-image input vs. fine-tuning data scale.
2. While MIS and MIRage are presented as key contributions, the fine-tuning pipeline itself largely follows a standard SFT paradigm. There is limited method innovation in model architecture or training strategy, which somewhat reduces the technical novelty of the work.
3. The MIS dataset appears unbalanced across categories, with nearly half of the samples belonging to Illegal Activity, while categories such as Self-Harm, Privacy, and Erotic account for less than 7%.

**Questions:**

1. How is the “Safety CoT” annotation structured—does the model generate step-by-step reasoning before the final refusal, or is it a single concatenated response?
2. How will the number of general QA samples used in MIRage affect models’ general performance?
3. The results in Table 4 show very high reasoning scores (RSR ~100%). Is it because after finetuning, the model has a very strong reasoning ability? Then what is the finetuned models’ performance on general multi-image reasoning datasets (e.g., ScienceQA) compared with other baselines?
4. In Table 5, after fine-tuning with MIRage, the model’s accuracy on MSS-Safe decreases by about 12%. Could the authors analyze the reason behind this drop?
5. Will the full dataset and finetuned models be released?

---

> ### Author Response · Authors · 2025-11-18
> **Official Comment by Authors (1/2)**
>
> We sincerely thank the reviewer for the thoughtful and thorough comments on our paper. We believe your feedback will improve the paper.
>
> ---
> > W1: The “Safety CoT” template is interesting but not well formalized or analyzed—it’s unclear how much of the improvement comes from reasoning versus simply from data diversity. The authors should include ablations to isolate the impact of Safety CoT vs. multi-image input vs. fine-tuning data scale.
>
> Thanks for your valuable question.
> - **We have presented ablation study in Tables 11 and 12**
> We conducted ablation studies on the MIRage label construction in Tables 11 and 12. The variant + MIRage w/ “I’m sorry” corresponds to the version without safety CoT reasoning. As shown in the results, this variant exhibits a notable decline in general performance (59.86 compared to 61.30 for MIRage) and performs worse on challenging safety benchmarks that require visual reasoning, further demonstrating the effectiveness of incorporating safety CoT annotations.
>
> ---
>
> >W2: While MIS and MIRage are presented as key contributions, the fine-tuning pipeline itself largely follows a standard SFT paradigm. There is limited method innovation in model architecture or training strategy, which somewhat reduces the technical novelty of the work.
>
> Our work does not aim to propose a new or superior training algorithm; instead, it focuses on enhancing model safety through strengthened visual reasoning capabilities. Our analysis in Section 2 highlights that existing methods face critical bottlenecks in balancing harmlessness and helpfulness, as well as in addressing challenging multimodal safety tasks.
>
> - **The Goal of MIS and MIRage**
> To address these challenges, our work introduces multi-image input data paired with safety CoT annotations. This design encourages models to perform safety-related cross-image visual reasoning, enabling them to infer and reason about safety rather than relying solely on surface-level perception. By enhancing visual reasoning, our approach improves both safety awareness and contextual understanding, achieving a better balance between helpfulness and harmlessness.
>
> - **A Novel yet Simple Dataset (MIS) to Resolve the Problem**
> Although we employ a standard SFT paradigm, our novel combination of multi-image inputs and safety CoT annotations effectively overcomes the bottlenecks of VLGuard and other baselines. This simple yet powerful dataset design substantially enhances the model’s safety performance by strengthening its visual reasoning capabilities.
> ---
>
> > W3: The MIS dataset appears unbalanced across categories, with nearly half of the samples belonging to Illegal Activity, while categories such as Self-Harm, Privacy, and Erotic account for less than 7%.
>
> Although some categories account for only about 7% of the entire dataset, the smallest category, Erotic, still contains 146 samples. This number is comparable to the least-represented categories in previous safety benchmarks (130 in VLSBench, 44 in MM-SafetyBench, 188 in MSSBench) and is sufficient to evaluate model performance within this safety domain. Moreover, we report detailed per-category evaluation results in Figure 9, further demonstrating the comprehensiveness of the MIS dataset.

---

> > ### Author Response · Authors · 2025-11-18
> > **Official Comment by Authors (2/2)**
> >
> > > Q1: How is the “Safety CoT” annotation structured—does the model generate step-by-step reasoning before the final refusal, or is it a single concatenated response?
> >
> > We design a safety CoT prompt that explicitly informs InternVL2.5-78B that the given query contains harmful intent. The prompt further instructs the model to first analyze the visual information in each image, then identify potential safety risks through cross-image reasoning, and finally generate a safe response. The detailed prompt is provided in Appendix H.2. The entire label is generated by the model in a single pass, without the need to concatenate multiple responses.
> >
> > ---
> >
> > > Q2: How will the number of general QA samples used in MIRage affect models’ general performance?
> >
> > - **We have conducted this experiment in Table 6**
> > As shown in the last two rows of Table 6, even without adding any additional general samples, MIRage achieves an average score of 60.53 on the five general benchmarks, which already surpasses the base model’s 60.47. Furthermore, by adding only 500 general QA samples, the average accuracy of MIRage increases to 61.30. This experiment demonstrates that the MIS training dataset can effectively achieve a balance between harmlessness and helpfulness, even without extra general data augmentation.
> >
> > ---
> >
> > > Q3: The results in Table 4 show very high reasoning scores (RSR ~100%). Is it because after finetuning, the model has a very strong reasoning ability? Then what is the finetuned models’ performance on general multi-image reasoning datasets (e.g., ScienceQA) compared with other baselines?
> >
> > We believe that the MIRage fine-tuned models exhibit enhanced visual reasoning capabilities.
> >
> > - **Superior performance on reasoning-intensive safety tasks**
> > In safety benchmarks that require reasoning to resolve harmful intent, MIRage outperforms all baselines. Both SIUO and MSSBench are designed to elicit unsafe responses by combining safe multimodal inputs to form unsafe intentions, which require strong reasoning ability for the model to provide safe answers. As shown in Tables 5 and 14, MIRage consistently surpasses the baselines on these two datasets, confirming that its improved safety stems from enhanced visual reasoning ability.
> >
> > - **Better general performance than the base model**
> > We evaluated MIRage on single-image, mixed single/multi-image, and multi-image general benchmarks. Notably, MuirBench and MMT-Bench include a large number of multi-image reasoning QA tasks, and MIRage achieves higher accuracy on both, demonstrating its strengthened reasoning capability. Actually, ScienceQA mainly involves single-image general tasks, our evaluations on MMStar and Q-Bench already provide sufficient evidence that MIRage generalizes well across different types of visual inputs.
> >
> > ---
> >
> > > Q4: In Table 5, after fine-tuning with MIRage, the model’s accuracy on MSS-Safe decreases by about 12%. Could the authors analyze the reason behind this drop?
> >
> > We also observed this phenomenon and believe it may result from **mis-evaluations by GPT-4o** on the Safe subset of MSSBench during the automatic evaluation process. Specifically, GPT-4o occasionally misjudges MIRage’s logically sound and safety-aware responses as overly conservative. **As illustrated in Figure 6 of the Appendix**, we provide a failure case from the MSSBench Safe set: the MIRage model gives a highly reasonable response, explicitly reminding that children should be cautious when playing tennis, which is in fact a correct classification. However, GPT-4o evaluated this output as an instance of over-conservatism and incorrectly marked it as a misclassification.
> >
> > ---
> > > Q5: Will the full dataset and finetuned models be released?
> >
> > Yes, we will release both the MIS training and testing datasets, as well as the MIRage fine-tuned models, once the paper is accepted.

---

### Official Review · Reviewer_wAXc · 2025-10-31

**Soundness:** 3
**Presentation:** 3
**Contribution:** 2
**Rating:** 6
**Confidence:** 3

**Summary:**

The paper addresses a safety gap in vision–language models (VLMs), where many safety-tuned VLMs identify content in harmful text or single-image inputs, but fail in combinations of two images with a neutral prompt. Following from this observation the authors make the (general) claim that current safety SFT methods (text-only safety SFT, multimodal SFT) do not actually teach (visual–visual–text) safety reasoning; either the models answer harmful composite queries, or they become over-conservative.
To this end, they introduce a dataset MIS (multi-image safety benchmark): 2,185 test items and ~4k SFT-style training items across 6 safety domains (illegal activity, violence, hate, self-harm, privacy, erotic), with three difficulty/style splits (easy, hard, real). Each example has two images + a detoxified/neutral instruction where, importantly, the unsafe intent only comes from the image–image relation.

The paper then SFTs several VLMs (e.g. InternVL2.5-8B) on these examples, where labels corresond to COTs describing the images conten and explain why the combination is unsafe. They evaluate the models against post-hoc or reconstructed VLGuard baselines on the same base model, showing large improvements on attack success rates (ASR), while performance on general multimodal benchmarks stays consistent.

**Strengths:**

- The paper addresses an interesting and important topic: safety gaps in multi-image VLMs, where it shows the benefits of COT reasoning to address these limitations.
- It introduces an interesting small dataset (MIS).
- The paper studies an interesting safety-loophole in in VLMs using two neutral images + text; in this scope it is well-framed, and building on existing work where neutral text and single images can still produce unsafe outputs.
- Overall, the experiments and evaluations are thorough (including training several VLMs) include several safety baselines, public safety datasets (FigStep, MSSBench, SIUO), general benchmarks to check utility, including ablations.
- The paper shows that multi-modal benchmark performance does not drop as it shows it is the case in other safety methods.
- The training method is simple: standard SFT on the MIS dataset.

**Weaknesses:**

The paper shows several weaknesses hampering generalizability of the approach aimed at addressing "a significant bottleneck in the safety capabilities of existing safeguarding methods" [line 045]:

- The focus is solely on two images + text (appears limited to generalize this claim)
- Along this line the datasets size also appears limited, especially for the smaller categories (e.g. self-harm, privacy), with only a few hundred images.
- The method is only evaluated on internally trained models and not compared against existing safety-tuned VLMs, such as [1,2].
- Given the somewhat narrow focus, I find the presentation too strong at times, e.g. "existing SFT methods are insufficient to provide effective defenses.” [line 100], “reveal a significant bottleneck in the safety capabilities of existing safeguarding methods.”, “the cause of safety bottlenecks can primarily be attributed to (i) composition of SFT inputs and (ii) construction method of SFT labels.” [line 101]


[1] https://arxiv.org/abs/2406.05113
[2] https://arxiv.org/abs/2406.12030

**Questions:**

Can you run MIS on at least one released safety VLM?
How does the method perform when we use more than two images?

**Details Of Ethics Concerns:**

As the dataset contains harmful images, it should be well-documented during release.

---

> ### Author Response · Authors · 2025-11-18
>
> Thanks for your valuable feedback, which greatly contributed to improving the paper!
>
> ---
> > W1.1: The focus is solely on two images + text (appears limited to generalize this claim)
> - **We consider this to be a core contribution of our work.**
> Inspired by the findings in Section 2, we aim to improve model safety by enhancing its reasoning capability. However, we observed that the input images used in VLGuard-R are too simple, making it difficult to construct high-quality safety CoT labels. To address this issue, we propose the MIS dataset, where harmful intentions naturally emerge through cross-image reasoning—merely combining two individually safe images can create a harmful scenario. Our paper demonstrates that, under this design, complex multi-image inputs are sufficient to construct meaningful cross-image reasoning settings and enable the creation of high-quality safety CoT annotations.
>
> - **We also evaluate MIRage against other baselines on a variety of single-image multimodal safety tasks and general-purpose multimodal tasks.**
> Results in Tables 5 and 6 show that, although MIRage is trained exclusively with multi-image inputs, its enhanced visual reasoning capability allows it to achieve strong performance even on single-image tasks, across both safety-related and general benchmarks.
>
> ---
>
> > W1.2 Along this line the datasets size also appears limited, especially for the smaller categories (e.g. self-harm, privacy), with only a few hundred images.
> - **The distribution and sample counts of specific safety categories shown in the paper correspond to those of our MIS test dataset**
> Although some categories account for only about 7% of the entire dataset, the smallest category, Erotic, still contains 146 samples. This number is comparable to the least-represented categories in previous safety benchmarks (130 in VLSBench, 44 in MM-SafetyBench, 188 in MSSBench) and is sufficient to evaluate model performance within this safety domain. Moreover, we report detailed per-category evaluation results in Figure 9, further demonstrating the comprehensiveness of the MIS dataset.
>
> ---
>
> > W1.3 & Q1: The method is only evaluated on internally trained models and not compared against existing safety-tuned VLMs, such as [1,2].
>
> - **We did not limit our comparisons to internally trained models;** we also evaluated multiple safety-tuned VLMs, as shown in Figure 4 and Table 15.
> For Qwen2-VL-7B-Instruct, we tested both the VLGuard and Textual SFT versions provided by VLSBench. In addition, following the VLSBench setting, we fine-tuned MiniCPM-V 2.6 and LLaVA-OneVision-7B using the same VLGuard and Textual SFT protocols. Across these settings, MIRage consistently achieved the best performance on various tasks, demonstrating its robustness in both safety and visual reasoning.
>
> - **We extended our evaluation to two publicly available, safety-tuned VLMs: Qwen2-VL-7B-Instruct finetuned on SPA-VL and SafeRLHF-V [1], respectively, using the MIS-easy and MIS-hard subsets.**
> As shown in the table below, these safety-tuned VLMs also fail to achieve near-zero ASR on the MIS test set, highlighting both the challenging nature of MIS and the existing performance gap in current models for multi-image safety tasks.
>
>     |Methods|MIS-easy (ASR ↓)|MIS-hard (ASR ↓)|
>     |--|:--:|:--:|
>     |Qwen2-VL-7B-Instruct|90.03|89.41|
>     |+ Textual SFT|36.84|42.35|
>     |+ VLGuard-R|17.13|22.16|
>     |+ SafeRLHF-V|68.56|70.30|
>     |+ SPA-VL|29.20|30.06|
>     |+ MIRage|1.55|0.78|
>
> ---
>
> >W1.4: Given the somewhat narrow focus, I find the presentation too strong at times, e.g. "existing SFT methods are insufficient to provide effective defenses.” [line 100], “reveal a significant bottleneck in the safety capabilities of existing safeguarding methods.”, “the cause of safety bottlenecks can primarily be attributed to (i) composition of SFT inputs and (ii) construction method of SFT labels.” [line 101]
> - **"existing SFT methods are insufficient to provide effective defenses"**
>     We respectfully disagree that this statement is overly strong. The results in Table 1 show that the performance of existing methods on MSSBench and SIUO is significantly worse than on FigStep. This clearly indicates that current approaches struggle with such challenging tasks.
> - **"reveal a significant bottleneck in the safety capabilities of existing safeguarding methods."**
>     Thanks for your suggestion. We have adopted more appropriate and softer wording in the revised version.
> - **"the cause of safety bottlenecks can primarily be attributed to (i) composition of SFT inputs and (ii) construction method of SFT labels.” [line 101]"**
>     We respectfully stand by this statement, as the Discussion and Findings throughout Section 2 consistently support the two points we highlight here. While we are confident in this conclusion, we have revised the wording for greater precision and to avoid any misinterpretation.
> ---
> # Reference
> [1] SafeRLHF-V, Ji, Jiaming, et al.

---

### Official Review · Reviewer_SD7u · 2025-10-31

**Soundness:** 2
**Presentation:** 3
**Contribution:** 3
**Rating:** 4
**Confidence:** 4

**Summary:**

This paper diagnoses a key failure in existing safety fine-tuning for Vision-Language Models: methods either fail to stop harmful outputs or become "over-conservative," refusing to answer benign questions. The authors attribute this to a "safety visual reasoning gap," where models can't reason about harmful intent implied by (even safe) visual contexts. To solve this, they introduce the Multi-Image Safety (MIS) dataset, a novel, synthetically-generated dataset featuring multi-image inputs paired with safety Chain-of-Thought labels. Fine-tuning on this dataset (MIRage) is shown to effectively reduce attack success rates on their new benchmark while improving performance on general VLM tasks.

**Strengths:**

1. The MIS dataset is the first to focus on multi-image safety scenarios, a necessary and complex domain. The proposed MIRage, which uses safety CoT labels to teach reasoning rather than just refusal, is an effective solution to the problem. The dataset can be a useful resource to the community.
2. The method achieves a good balance of helpfulness and harmlessness. The results in Table 4 (near-0% ASR on their benchmark) and Table 6 (a slight increase in average accuracy on general benchmarks) show a good trade-off between safety and helpfulness.
3. The paper is well-written and easy to understand.

**Weaknesses:**

1. In Finding 1 and Discussion 1, the authors state that “fine-tuning models on such data leads to over-prudence on visual features, causing the model to reject benign visual inputs.” However, the exact experimental configuration for VLGuard-P is unclear. In the original VLGuard post-hoc fine-tuning setup, 5k general helpfulness samples were included. Therefore, two concerns arise: (1) if this experiment did not follow the same configuration, the conclusion may not be directly comparable; or (2) if it did follow the exact setup, then the total training set would include 6k benign image-text pairs—three times more than the 2k unsafe pairs—making “over-prudence on visual features” an unconvincing explanation.
2. Both the MIS training data and the primary test data (MIS-easy/hard) are generated using a T2I model (Stable Diffusion 3.5). And many of the benchmarks are synthetic such as FigStep, MM-Safety. It's unclear how it generalizes to real-image safety benchmarks. The "MIS-real" split, which would validate generalization, is too small to be conclusive.
3. The paper focuses exclusively on multimodal safety, providing zero evaluation on standard text-only safety benchmarks. A primary VLM attack method is still text-only jailbreaking, and it's unknown if this specialized visual safety fine-tuning has inadvertently created new vulnerabilities or "catastrophically forgotten" textual safety alignments.

**Questions:**

In Table 10, the inference latencies such as tokens/s change. Since the model is only fine-tuned on the MIRage dataset without a modification to the architecture, why would that change?

---

> ### Author Response · Authors · 2025-11-17
>
> Thank you for your great efforts on the review and constructive comments.
> > W1
> - **Clarification of our Experimental Setting**
> For the results in Table 1, as stated in our supplementary material, *LLaVA + VLGuard-M* and *LLaVA + VLGuard-P* are directly adopted from the **official VLGuard releases**. Therefore, the fine-tuning data for *LLaVA + VLGuard-P* indeed include the additional 5k general helpfulness samples you mentioned.
> For the remaining VLGuard-based variants reported in the table, since the **VLGuard paper did not release** the specific 5k general helpfulness samples, we strictly followed the setting in VLSBench [1] and fine-tuned the models using only the 2k unsafe and 1k safe samples publicly available from VLGuard.
>
> - **Fairness of Our Chosen Configurations**
>     1. Comparing the three VLGuard variants on the LLaVA backbone, we observe that both *VLGuard-M* and *VLGuard-P* show a **notable drop in general task performance**. In contrast, our constructed *VLGuard-R*, despite not including the extra 5k general samples, achieves significant safety improvement without harming general ability. This supports our **Discussion 1** (VLGuard tends to be over-conservative) and **Finding 2** (reasoning-based labels mitigate the loss of general capability).
>     2. Furthermore, comparing *VLGuard-P* and *VLGuard-R* on the Qwen and InternVL backbones demonstrates that, under the **same fine-tuning inputs**, models trained with reasoning-based labels (*VLGuard-R*) achieve superior general performance compared to those trained with rejection-based labels (*VLGuard-P*). This comparison is entirely fair, as the only variable is the label design, which is exactly the factor we aim to study.
>
> - **Discussion on the Inclusion of the 5k General Samples (Table 6)**
> Since the original VLGuard paper did not specify the exact 5k general samples, we randomly sampled 5k general QA examples from our general training set and added them to the *VLGuard-R* dataset. Comparing *VLGuard-R†* and *VLGuard-R‡*, we find that adding more general data does not further improve overall performance—it only reduces the model’s over-prudence.
> In contrast, *MIRage* consistently achieves better general capability across different input ratios, validating the effectiveness and robustness of our method.
>
> ---
> > W2
>
> - **Evaluation on real-image safety datasets**
> We further compared MIRage with other baselines on several real-image safety datasets, such as MSSBench and SIUO (as shown in Table 5). These datasets are designed to elicit harmful responses by combining safe images and texts to construct unsafe intentions, and importantly, **all images are real** rather than synthetic. Moreover, **MSSBench is a large-scale dataset** — the chat split we evaluated contains 600 real image–text pairs.
> The strong performance of MIRage on both MSSBench and SIUO demonstrates that its acquired safety capabilities generalize effectively to real-world images.
>
> - **Reliability of MIS-real dataset**
> As shown in Table 4, the results indicate that different models exhibit consistent performance across MIS-real, MIS-easy, and MIS-hard, confirming the reliability and validity of our MIS-real dataset, even though it contains only 100 samples.
> ---
> > W3
>
> - **Why we did not evaluate on text-only tasks**
> Since our method primarily enhances safety through improved visual reasoning, our evaluations naturally focus on multimodal tasks. The benchmark suite includes general QA datasets, basic multimodal safety datasets, and challenging safety benchmarks such as MSSBench and SIUO, which are designed to test the model’s ability to reason about safety in complex visual contexts.
>
> - **Additional evaluation on text-only jailbreaks**
> To address your concern, we further followed the evaluation setting in ETA [2] and conducted additional experiments on AdvBench by adding adversarial suffixes to assess the MIRage model’s robustness against text-only jailbreak attacks. As shown in the Table below, the fine-tuned MIRage model **did not suffer from catastrophic forgetting**; instead, it successfully generalized multimodal safety awareness to the text-only domain, improving the model’s safety against purely textual jailbreak attempts. We have included the corresponding experimental setup and discussion in the revised manuscript.
>    |Model (ASR ↓)|AdvBench + adversarial suffix|
>    |-|:-:|
>    |InternVL2.5-8B|21.34|
>    |+ MIRage|8.07|
> ---
> > Q1
>
> Thank you for your valuable comment. Except for the Qwen2-VL-7B, which showed relatively higher inference latency on MSSBench, all other results indicate that the inference latency of the MIRage-tuned models is comparable to their base VLMs. After re-evaluating the Qwen2-VL-7B and Qwen2-VL-7B+MIRage models on MSSBench, we found that the previous latency difference was caused by hardware issues. We have revised the manuscript. We sincerely appreciate your correction.
>
> ---
> # Reference
> [1] VLSBench, Hu, Xuhao, et al.
>
> [2] ETA, Ding, Yi, et al.

---

> > ### Comment · Reviewer_SD7u · 2025-11-25
> >
> > Thank you for your clarification and additional experiments. Hope these changes can be incorporated in the final version. I'm raising my score to 6.

---

> > > ### Author Response · Authors · 2025-11-28
> > >
> > > Thank you for your response and for increasing your rating. We truly appreciate your valuable insights, which have helped us improve the quality of our work. We will incorporate the suggested experiments and discussion in the revised version. Thank you again for your support of our paper!

---

### Author Response · Authors · 2025-12-04
**Rebuttal Summary for AC**

Dear AC,

We sincerely thank you for the time and effort you dedicated to the review, and we offer our profound respect for your commitment to academic fairness under these special circumstances. This document serves as our summary for the rebuttal phase, encompassing our responses to the reviewers' comments and further updates, which we hope facilitates your quick understanding.


**Initial reviews and acknowledged strengths**

The submission received the following initial scores:

Reviewers gNSB, wAXc, PDkF: Score of 6

Reviewers SD7u, v3WZ: Score of 4

While the scores varied, we are encouraged that all reviewers identified the MIS dataset as a significant contribution and MIRage successfully improves safety without the common downside of degrading general model performance. The reviewers (PDkF, SD7u, wAXc, gNSB) praised our paper systematic diagnosis of safety bottlenecks in current VLMs for multi-image scenarios. And reviewers (wAXc, v3WZ, PDkF) commended the thoroughness of the evaluation and the method's ability to generalize.

**Summary of rebuttal efforts**

We try to address the reviewers' concerns throughout the rebuttal phase, the efforts are summarized as follows:

* We additionally evaluate MIRage on **text-only jailbreak benchmark** Advbench, demonstrating that its safety capabilities also transfer to purely textual safety scenarios.
* On MIS, we compare against **more publicly available safety-tuned VLMs** beyond VLGuard and Textual SFT, including Qwen2-VL-7B-Instruct finetuned on SPA-VL and SafeRLHF-V. These results further highlight the limitations of existing methods in complex multimodal settings.
* We report the performance of both **7B-scale and 72B/78B-scale models under prompt-based customized Safety-CoT**. We find that such inference-time methods fail to achieve satisfactory ASR on MIS regardless of model size, and they additionally degrade general performance.
* We elaborate on the **contributions of our work**, explaining the bottlenecks faced by current safety-finetuning approaches. In contrast, MIRage leverages complex direct inputs and safety-CoT labels in the MIS training set to enhance visual reasoning, which in turn strengthens its safety behavior.
* We compare the safety-category **data scale** of MIS test sets with existing VLM safety benchmarks, showing that MIS offers a more comprehensive evaluation of multimodal safety.


**Positive feedback and current status**

* Reviewer SD7u (Score 4 -> Score 6): Following our clarification and additional experiments, this reviewer stated "the changes can be incorporated in the final version" and appreciated the reply, raising the score to 6.
* Reviewer gNSB (Score 6): Maintain the rating that the reviewer already leans towards accepting the paper.
* Reviewer wAXc (Score 6): While this reviewer has not replied, the reviewer leans towards accepting the paper.
* Reviewer PDkF (Score 6): While this reviewer has not replied, the reviewer leans towards accepting the paper.
* Reviewer v3WZ (Score 4): We conducted detailed experiments and provided thorough responses to the reviewer’s concerns raised during the review stage, most of which were acknowledged positively. We also carried out additional experiments addressing the new issues raised during the discussion phase. Specifically, for the 72B and 78B model sizes, we reported the safety and utility performance under prompt-based Customized Safety-CoT, further validating the adaptability of our findings. Moreover, the reviewer noted during the review stage: “I would be likely to support acceptance if this concern is adequately addressed and the claims are adjusted based on the results with the adapted prompt.” Our expanded experiments and clarified claims directly provide evidence to address the review’s concern, and we believe the experiments on different size models further enhance the comprehension of our experiments.


Thank you again for your time and effort. Your dedication contributes to making ICLR a more open, fair, and equitable conference.

Sincerely,

Submission13706 Authors

---

### Meta-Review · Area_Chair_uwdb · 2026-01-07

**Summary:**

The paper studies a more complex safety setup than finetuning VLGuard data alone, since models are prone to over-refusal on benign data and struggle with complex safety situations where compositional dependencies between text and image can lead to an unsafe request. The paper introduces a dataset (MIS) that is used for both fine-tuning of pre-trained MLLMs and also evaluating them. Experimental evaluation shows that existing models can be easily jailbroken using the data, but a model fine-tuned using the data is resistant to these kinds of attacks.

Reviewers raised concerns which include:
(1) The described method appears to be essentially common fine-tuning / VLGuard with different data.
(2) The main results seem to only use one model, even if there are some results on other models in the appendix.
(3) Minor typographical errors.
(4) Significant performance gap between non-finetuned and finetuned model, despite the same model providing the finetuning labels, suggesting prompting issues.
(5) It is unclear how much improvement comes from data versus CoT changes; ablations would help to clarify.
(6) The finetuned model appears to decrease in accuracy (Table 5) without explanation.
(7) Unclear whether dataset and model will be released, and concerns around dataset balance and size.
(8) Changing latencies without architectural changes.

**Reviewer Concerns:**

The authors provided rebuttal responses addressing (1-2) and (4); while the paper is not heavily modified in response to reviewer critique, all points are acknowledged and explained, and may have been influential in the discussion period. All typographical errors raised in (3) appear to be corrected. The authors clarify with additional ablations towards (5), and explain the loss in performance in (6). Authors will release data (7) upon acceptance. (8) was explained as a hardware error and corrected in the manuscript.

**Reviewer Scores:**

gNSB would maintain their score (6) after the discussion period, as indicated by their comments.
v3WZ expressed concerns about prompt methods and consistency, discussed with reviewers during the rebuttal, and did not indicate satisfaction sufficient to change the score; I would expect maintaining their score (4).
PDkF and wAXc would likely maintain their scores (6).
SD7u remarked that they would raise their score to 6 following discussion.

---

### Decision · Program_Chairs · 2026-01-26

Accept (Poster)